# Two-dimensional metal–organic framework for post-synthetic immobilization of graphene quantum dots for photoluminescent sensing
You-Liang Chen[1], Darwin Kurniawan[2], Meng-Dian Tsai[1], Jhe-Wei Chang[1], Yu-Na Chang[1], Shang-Cheng Yang[1], Wei-Hung Chiang [2,3,4] & Chung-Wei Kung [1] ✉

Immobilization of graphene quantum dots (GQDs) on a solid support is crucial to prevent GQDs from aggregation in the form of solid powder and facilitate the separation and recycling of GQDs after use. Herein, spatially dispersed GQDs are post-synthetically coordinated within a two-dimensional (2D) and water-stable zirconium-based metal–organic framework (MOF). Unlike pristine GQDs, the obtained GQDs immobilized on 2D MOF sheets show photoluminescence in both suspension and dry powder. Chemical and photoluminescent stabilities of MOF-immobilized GQDs in water are investigated, and the use of immobilized GQDs in the photoluminescent detection of copper ions is demonstrated. Findings here shed the light on the use of 2D MOFs as a platform to further immobilize GQDs with various sizes and distinct chemical functionalities for a range of applications.

Graphene quantum dots (GQDs) are nanometer-sized graphene layers with delocalized π electrons at the centre and versatile oxygen-containing functional groups at the edges[1]. With the unique features such as the exceptional solution processability, excellent luminescent performance, tunable luminescent property, and photostability[2,3], GQDs have been applied for various applications including chemical sensors[4], catalysis[5], and drug delivery[6]. However, the aggregation of GQDs can lead to the fluorescent quenching, which strongly limits the direct applications of GQDs in the solid form[7]. In addition, the high solubility of GQDs in aqueous media results in the difficulty in separating and recycling GQDs after use[8]. The immobilization of GQDs on a solid support, e.g., mesoporous silica[9] or polymer[10], has thus become an appealing strategy to overcome these challenges of GQDs aiming for practical applications.

As a class of nanoporous materials, metal–organic frameworks (MOFs) have attracted considerable attention and been utilized in various applications owing to their fascinating characteristics such as the exceptionally high surface area, interconnected pore structure and diverse chemical functionality[11–17]. These features have thus made MOFs become appealing solid supports for GQDs[18]. The high surface area and porosity of MOFs should allow the immobilization of spatially separated GQDs with a high loading, and the interconnected pores are expected to render GQDs fully accessible to the reactant during the catalysis or sensing. The development of water-stable group(IV) metal-based MOFs[19,20], such as zirconium-based MOFs (Zr-MOFs), has further opened up the opportunity to design such MOF-GQD composites aiming for the applications in aqueous environments. Numerous efforts have thus been made to incorporate GQDs in MOFs in the literature[18,21–26]. It should be noticed that since harsh synthetic conditions are usually required to form GQDs, unlike other materials such as inorganic quantum dots and carbon quantum dots (CQDs)[27], the in-situ synthesis of GQDs in the presence of as-prepared MOFs is fairly challenging. Thus, in most previous studies, encapsulation was used to incorporate GQDs within MOFs[18,21–25]. By synthesizing the MOF in the solution containing GQDs, spatially separated GQDs can be encapsulated within the obtained MOF crystals, but such approaches usually result in the random and dispersed distribution of GQDs in the composite. Post-synthetic modification (PSM)[28,29] may provide another option to immobilize GQDs in MOFs. Since both MOFs and GQDs are synthesized separately before preparing the composite, PSM is supposed to provide more synthetic flexibility. For example, in our previous work, the post-synthetic immobilization of GQDs within a MOF was reported for the first time[26]. GQDs with a relatively small average size (3.1 nm) and a three-dimensional (3D) Zr-MOF with larger mesopores were used, which could

[1]Department of Chemical Engineering, National Cheng Kung University, Tainan City, Taiwan. [2]Department of Chemical Engineering, National Taiwan University of Science and Technology (NTUST), Taipei City, Taiwan. [3]Sustainable Electrochemical Energy Development (SEED) Center, NTUST, Taipei City, Taiwan. [4]Advanced Manufacturing Research Center, NTUST, Taipei City, Taiwan. ✉e-mail: cwkung@mail.ncku.edu.tw

lead to the uniform and spatially continuous immobilization of GQDs within the entire 3D MOF structure[26,30]. However, since the pore sizes of most MOFs are smaller than the sizes of most GQDs, such synthetic methods are barely generalizable; with GQDs larger than the MOF pore, the post-synthetic immobilization of GQDs would only occur on the external surface of MOF crystals, which is unfavorable for achieving a sufficiently high loading of GQDs.

Herein, a generalizable synthetic approach was demonstrated to post-synthetically immobilize GQDs within the entire structure of a MOF, by utilizing a two-dimensional (2D) Zr-MOF, ZrBTB (BTB = 1,3,5-tri(4-carboxyphenyl)benzene)[31], as the platform. This is the first study reporting the composite composed of a 2D MOF and GQDs. With the fully dispersed 2D molecular sheets of ZrBTB in the solution of GQDs, the spatial limitation originating from the pore size of 3D MOFs no longer exists during the immobilization of large GQDs. It is thus expected that GQDs with terminal carboxylic groups may coordinate on the terminal hydroxo/aqua ligands present on the six-connected hexa-zirconium node of ZrBTB to achieve the immobilization of GQDs, which is similar to the coordination of small ligands in Zr-MOFs *via* solvent-assisted ligand incorporation (SALI)[32,33]. As shown in Fig. 1, the immobilization of GQDs with three diverse sizes between 2 nm and 5 nm was demonstrated, and 2D MOF-GQD materials in the form of solid powder while preserving the luminescent characteristics of GQDs can be successfully synthesized.

## Results and discussion
### Materials characterization
Three GQD materials with diverse average sizes, named as "GQD-1", "GQD-2", and "GQD-3", were synthesized by utilizing microplasma at ambient conditions with the pyromellitic acid, chitosan, and 2-phenylphenol as precursors, respectively[34,35]; see experimental details and characterizations of GQDs in Supplementary Methods, Supplementary Notes 1–4, Figs. S1–S5, and Table S1. All GQD materials possess enriched carboxylic and hydroxyl groups useful for further PSM, and aqueous solutions of all of them are luminescent. However, no obvious luminescence can be observed from all of them in the form of solid powder. The immobilization of spatially separated GQDs on a solid support is thus required to prevent such an aggregation of GQDs that quenches their photoluminescence.

Solution-phase post-synthetic immobilization of these GQDs in ZrBTB, which possesses accessible –OH/–OH$_2$ ligands on its nodes[36,37], was then performed to prevent the aggregation of GQDs in the solid form; see experimental details in "Methods" section. It is worth mentioning that ZrBTB was synthesized with the use of benzoic acid as the modulator, and the benzoate-coordinated 2D MOF, ZrBTB-BA, was further treated with hydrochloric acid to remove all capped benzoate ligands[36], as evidenced by the ¹H nuclear magnetic resonance (NMR) data shown in Fig. S6. After the complete removal of benzoate, terminal –OH/–OH$_2$ ligands become accessible on each node of the resulting ZrBTB, which are expected for the

coordination of GQDs containing carboxylic groups on their surface. By fully dispersing the 2D ZrBTB in solutions of the three GQDs (Supplementary Note 5 and Fig. S7), GQDs could be immobilized on the 2D molecular sheet of ZrBTB. Powder X-ray diffraction (PXRD) patterns of ZrBTB, GQD-1-ZrBTB, GQD-2-ZrBTB, and GQD-3-ZrBTB are shown in Fig. 2a. Main diffraction peaks of the simulated ZrBTB[31] can be found in the experimental patterns of all MOF-based materials, indicating that the crystallinity of ZrBTB can be preserved after the installation of all GQDs. It is worth mentioning that the small diffraction peak located at 10.5° in the pattern of ZrBTB, which is attributed to the (002) plane indicating the spacing between 2D sheets of ZrBTB[31,38], shifts to ~10.0° in the patterns of all the three composites, which implies the expansion of the space between 2D MOF sheets after the incorporation of GQDs. N$_2$ adsorption–desorption isotherms were then collected to probe the porosity of each material. As shown in Fig. 2b, all the obtained isotherms exhibit a sharp N$_2$ uptake at low relative pressures and a hysteresis loop in the range of high relative pressures, which correspond to the micropores of the stacked ZrBTB molecular sheets and the mesoporosity originating from the space between particles of stacked 2D sheets, respectively[37,39]. Brunauer-Emmett-Teller (BET) surface areas of all MOF-based materials are listed in Fig. 2b. The BET surface area of ZrBTB was estimated as 310 m²/g, agreeing well with those reported previously[37,40]. It should be noted that owing to the 2D nature of ZrBTB, the solvent used during the activation of the MOF can significantly affect the stacking of these 2D MOF sheets and therefore the resulting BET surface area[41,42]; its reported BET surface areas range from 220 to 450 m²/g depending on the final solvent used before the activation[37,39,40,42]. After the immobilization of GQD-1, the resulting composite possesses a slightly higher BET surface area (360 m²/g), which should be attributed to the coordinated GQD-1 that further activated the already present micropores in ZrBTB. It is worth mentioning that ZrBTB can be fully dispersed as separated molecular sheets in the solution to allow the immobilization of large GQDs, but in the form of dry solid, these 2D sheets stack to each other with an inter-layer distance of ~0.7 nm[31]. Since GQDs are composed of planar graphene layers with large sizes rather than spherical particles, they could be immobilized on the 2D sheet of ZrBTB and show the orientation almost parallel to the stacked MOF sheets after drying, resulting in the slight expansion of the microporous space between stacked 2D sheets. Similar observations of increased BET surface area were also reported for ZrBTB after the post-synthetic coordination of small ligands[37,40]. But after the immobilization of GQD-2 and GQD-3 with larger sizes, BET surface areas of the obtained composites become 260 and 190 m²/g, respectively. Pore size distributions of all MOF-based materials were further extracted from the isotherms, and as revealed in Fig. S8, the main pore size of ZrBTB at ~1.2 nm can be observed for all MOF-based materials before and after installing GQDs.

X-ray photoelectron spectroscopy (XPS) spectra were used to examine the interaction between the immobilized GQDs and hexa-zirconium clusters of the 2D MOF; the shift of binding energy in XPS spectra after the

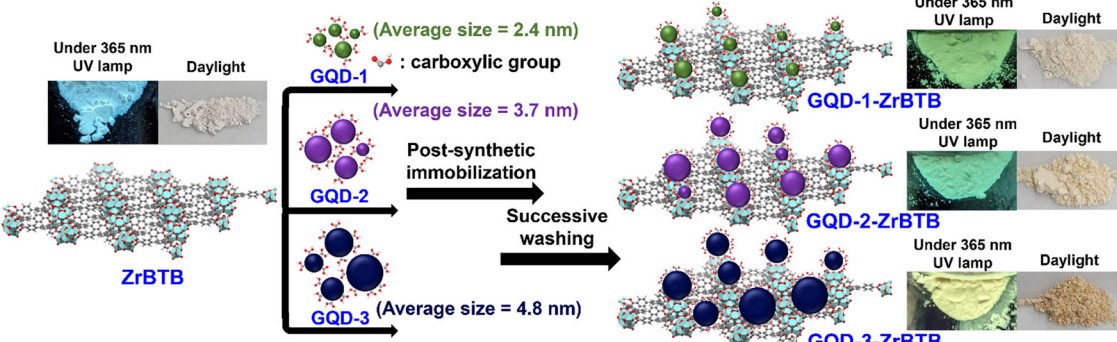

**Fig. 1 | Schematic representation for the post-synthetic immobilization of GQDs with various sizes on 2D ZrBTB MOF sheets.** Photographs of all MOF-based powders taken under the illumination at 365 nm and daylight (white light), respectively, are also shown.

**Fig. 2 | Materials characterization data of MOF-based materials. a** PXRD data, (**b**) $N_2$ isotherms, (**c**) Zr 3d XPS spectra, and (**d**) FTIR spectra of MOF-based materials. Locations of Zr $3d_{5/2}$ peaks are marked in (**c**).

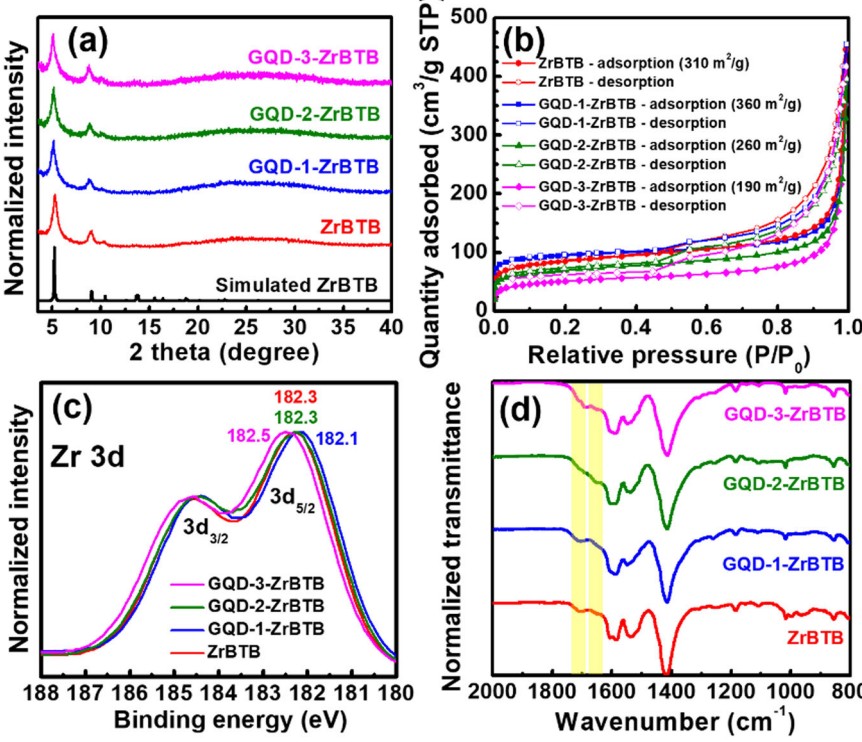

formation of coordination between organic ligands and metal ions has been widely observed in the fields of both organometallic complexes and MOFs[43–46]. In Fig. 2c, XPS data show that after the installation of GQD-1 and GQD-3 in ZrBTB, a negative shift and a positive shift of 0.2 eV can be observed for Zr 3d peaks, respectively, indicating that the immobilization of GQD-1 and GQD-3 makes the Zr atoms in hexa-zirconium nodes more electron-rich and electron-deficient, respectively. Similar observations have been reported previously for the coordination of small ligands in Zr-MOFs *via* SALI[46]; findings here thus suggest the successful coordination of GQD-1 and GQD-3 on the nodes of ZrBTB. The different directions of peak shifting may be attributed to the distinct sizes of GQDs. It has been reported that larger GQDs have lower lowest unoccupied molecular orbitals (LUMO) and thus become more electron-withdrawing compared to smaller GQDs[47]. Thus, the coordination of large GQD-3 makes the hexa-zirconium node of ZrBTB more electron-deficient, while the coordination of small GQD-1 that is more electron-donating results in the negative shift of Zr 3d peaks. However, such a peak shifting is not observable in the XPS data of GQD-2-ZrBTB. But from the survey and C 1s spectra of GQD-2-ZrBTB, the presence of GQD-2 in the composite was confirmed (Supplementary Note 6 and Fig. S9).

Fourier-transform infrared (FTIR) spectra of all MOF-based materials were then collected (Fig. 2d; see detailed discussions in Supplementary Note 7). The peak of terminal carboxylic groups located at 1710 cm$^{-1}$ in the FTIR spectra of all other materials, originating from those linkers that are not fully coordinated, becomes less obvious in the spectrum of GQD-2-ZrBTB. In addition, another peak located at 1657 cm$^{-1}$ (highlighted in yellow in Fig. 2d) can be clearly observed in the spectrum of GQD-2-ZrBTB, which should be associated with the linkage between the edged carboxylate group of ZrBTB and the amino group of GQD-2[48]. Both XPS and FTIR findings imply that the majority of GQD-2 should be immobilized on the uncoordinated carboxylic groups of linkers either present at structural defects in the 2D MOF sheet or on the edge of the sheet, rather than being coordinated on the nodes. Even though the majority of GQD-2 are not coordinated on the hexa-zirconium nodes of ZrBTB, the immobilization is still fairly firm, as evidenced by the photoluminescent data shown in Figure S10. Raman spectroscopic experiments of all MOF-based materials were attempted to further characterize the immobilized GQDs, but it was not successful owing

to the presence of strong Raman peaks from the pristine ZrBTB within the same region of the characteristic peaks of GQDs (Fig. S11).

The loading of GQDs in each composite was further determined by quantifying the mass fraction of zirconium in each material, and as discussed in detail in Supplementary Note 8, the mass fractions of GQDs in GQD-1-ZrBTB, GQD-2-ZrBTB, and GQD-3-ZrBTB are 12.8%, 10.3%, and 20.2%, respectively. Mass fractions of GQDs in all the three composites are higher than 10 wt%, which implies that GQDs are immobilized between the stacked 2D sheets of ZrBTB rather than solely attached on the external crystal surface of stacked 2D MOF sheets.

ZrBTB is composed of flower-like stacked 2D sheets[36,37], and such a 2D morphology is still preserved after the immobilization of all GQDs (Fig. S12). However, some dark particles attaching on the 2D sheet can be observed in the three MOF-GQD composites (Fig. S13). High-angle annular dark-field scanning transmission electron microscopy (HAADF-STEM) images were then collected to probe the GQDs immobilized on the crystalline 2D MOF sheet. As revealed in the representative HAADF-STEM images shown in Figs. S14 and 3a–c, several bright particles appear on the 2D MOF sheets of all composites. Size distributions of these particles were then estimated from the images. For example, particles in GQD-3-ZrBTB have an average size of 4.41 ± 1.1 nm (Fig. 3d), which agrees with the average particle size of the pristine GQD-3 shown in Fig. S3. Size distributions of particles in GQD-1-ZrBTB and GQD-2-ZrBTB also agree well with those of the corresponding GQDs (Fig. S15). Results here clearly confirm the successful immobilization of GQDs with various sizes onto the 2D ZrBTB sheets. In addition, both GQDs and clear lattice fringes can be observed in the HAADF-STEM image. As revealed in Fig. 3e, f, the spatial arrangement of bright spots appearing in the lattice fringes is consistent with that of the hexa-zirconium clusters present in the crystalline structure of ZrBTB. Furthermore, the obtained distance between adjacent spots (1.77 nm) agrees well with the node-to-node distance of the simulated ZrBTB (around 1.8 nm)[31]. Such lattice fringes of ZrBTB are also observable in the images of GQD-1-ZrBTB and GQD-2-ZrBTB (Fig. S15). It is worth noting that since all GQDs have much larger sizes compared to the aperture size of the 2D MOF and all three kinds of GQDs have their own ranges of particle size distributions (Fig. S3), it is not viable to observe the immobilized GQDs exactly correlated to the lattice fringes of ZrBTB. From Fig. 3a–c, some

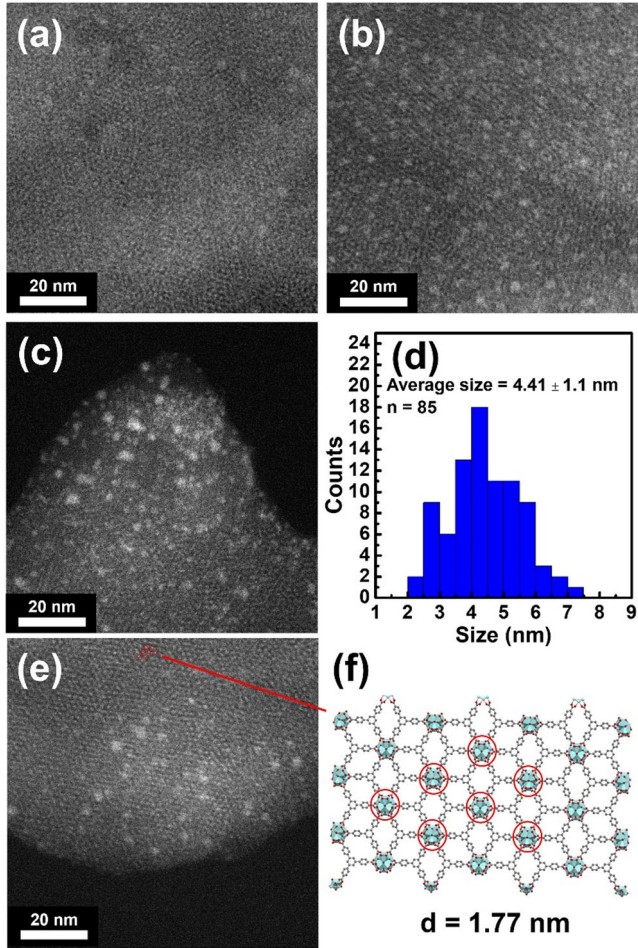

**Fig. 3 | HAADF-STEM data of MOF-based materials.** Representative HAADF-STEM images (**a**) GQD-1-ZrBTB, (**b**) GQD-2-ZrBTB, and (**c**) GQD-3-ZrBTB. **d** Particle size distribution of GQDs in GQD-3-ZrBTB. **e** HAADF-STEM image of GQD-3-ZrBTB showing both immobilized GQDs and lattice fringes of MOF. **f** Structure of ZrBTB with one layer.

aggregations of GQDs on the 2D MOF sheet can also be found, especially more obvious in GQD-3-ZrBTB.

### Photoluminescent (PL) properties of materials

Emission spectra of all GQD solutions and the suspensions of MOF-based materials were then collected under an excitation at 370 nm (see absorption and excitation spectra in Figs. S16 and S17 and corresponding discussions in Supplementary Notes 9 and 10). As shown in Fig. S18, emissions of various GQD materials at distinct characteristic wavelengths can be clearly observed both before and after immobilizing them in ZrBTB, but their emission spectra exhibit quite different characteristics. Although the sizes of GQDs follow the order of GQD-3 > GQD-2 > GQD-1, the wavelengths of maximum emission of these GQD solutions do not follow this order. It has been reported that a higher content of oxygen-containing groups can cause a redshift in the PL emission of GQDs[49,50]. Since GQD-1 possesses an oxygen-to-carbon ratio of 1.46, which is much higher than those of GQD-2 (0.42) and GQD-3 (0.32) (Table S1), despite having the smallest average particle size, GQD-1 exhibits the longest emission wavelength among all the three kinds of GQDs. On the other hand, doping graphitic nitrogen into GQDs can result in a red-shift PL behavior, while both pyridinic nitrogen and pyrrolic nitrogen dopants can induce a blueshift in the PL emission[49]. From Fig. S2d, it can be observed that GQD-2 contains various kinds of nitrogen dopants except graphitic nitrogen, leading to a strong blueshift in its emission spectrum. These results imply that surface functionalities could

provide a stronger impact in the PL emission of GQDs than the particle size, especially when the difference in particle size is relatively small, *i.e.*, around 1-2 nm. In addition, Fig. S18a also reveals that under the same concentration, the emission intensities of GQD solutions follow the order of GQD-1 > GQD-2 > GQD-3, which is consistent with the order of their oxygen-to-carbon ratios listed in Table S1. This observation should be attributed to the fact that a high content of surface functionalities in GQDs can prevent the π-π stacking-induced photon reabsorption and nonradiative energy transfer between GQDs, reducing the PL quenching phenomenon[51]. After the immobilization of GQDs in ZrBTB, as shown in Fig. S18b, both the wavelengths of maximum emission and PL intensities in general follow the trend of GQD solutions, but some obvious peak shifts can be observed.

All spectra were thus normalized to allow a fair comparison in peak shifts, and the results are plotted in Fig. 4a, b. Solutions of GQD-1, GQD-2, and GQD-3 show the wavelengths of maximum emission at 536 nm, 450 nm, and 470 nm, respectively. After immobilizing GQDs in MOF, the maximum emissions of GQD-1 and GQD-3 shift to 504 nm and 490 nm, respectively. Since the aggregation of GQDs can only significantly alter the intensity of PL emission rather than the wavelength of maximum emission[52], such peak shifts should not be attributed to the difference in the degree of GQDs aggregation. It should be noticed that increasing the electron density of GQDs can narrow the band gap, which results in the redshift of the PL emission[53]. According to the XPS data shown in Fig. 2c, the immobilization of GQD-1 and GQD-3 in ZrBTB can render the hexa-zirconium nodes of MOF more electron-rich and electron-deficient, respectively, which indicates that GQD-1 and GQD-3 become more electron-deficient and electron-rich after immobilization, respectively. Therefore, significant blueshift of emission for GQD-1 and redshift of emission for GQD-3 can be observed after the immobilization. On the other hand, since the majority of GQD-2 were immobilized on the uncoordinated linkers instead of the nodes (see FTIR and XPS results), no peak shift can be observed in its emission spectrum after the immobilization. Time-resolved PL measurements were also performed for all GQD solutions and suspensions of MOF-GQD materials. As shown in Fig. S19, the suspensions of GQD-1-ZrBTB, GQD-2-ZrBTB, and GQD-3-ZrBTB possess around 70%, 62%, and 58% of the corresponding PL lifetimes of homogeneous GQD solutions, but all PL lifetimes are in the same order of magnitude within a few nanoseconds. Findings here suggest that with the chemical immobilization of GQDs on the 2D MOF sheets, the aggregation of more GQDs in the resulting MOF-GQD composites may occur compared to that in homogeneous solutions, but such an aggregation is not quite significant. Emission spectra of all the three MOF-GQD solid powders were also collected. As revealed in Fig. 4c, PL properties of all GQDs installed in the MOF can still be observed in the form of dry solid, suggesting the successful immobilization of GQDs that are free from serious aggregation in ZrBTB. The PL property of the immobilized GQDs is also highly stable in water; no change in the emission intensity can be observed after suspending the GQD-1-ZrBTB in water for 1 h (Fig. 4d).

### Detection of copper ions

Since GQDs have been widely utilized in the PL detection of metal ions[4,54,55], as a demonstration, the composite with the strongest emission here, GQD-1-ZrBTB, was selected for testing its applicability in the PL detection of $Cu^{2+}$ ions (see experimental details in Supplementary Methods). As shown in Fig. 4e, obvious luminescent quenching of GQD-1-ZrBTB can be observed after adding $Cu^{2+}$ ions. According to the PL intensities recorded at 504 nm, the calibration curve for detecting $Cu^{2+}$ ions was obtained, with a linear range of 2.5–100 μM and a Stern–Volmer constant of 24200 $M^{-1}$ (Fig. 4f). The suspension of the pristine ZrBTB and the homogeneous solution of GQD-1 were also subjected to the PL detection of $Cu^{2+}$ ions. As shown in Fig. S20, no sensing response can be observed for the pristine ZrBTB, while the PL intensity of the GQD-1 solution can be significantly quenched after adding $Cu^{2+}$ ions. This finding indicates that GQD-1 is the active material responsible for the PL sensing of $Cu^{2+}$. It is worth mentioning that even though the homogeneous solution of GQD-1 seems to show a better sensing

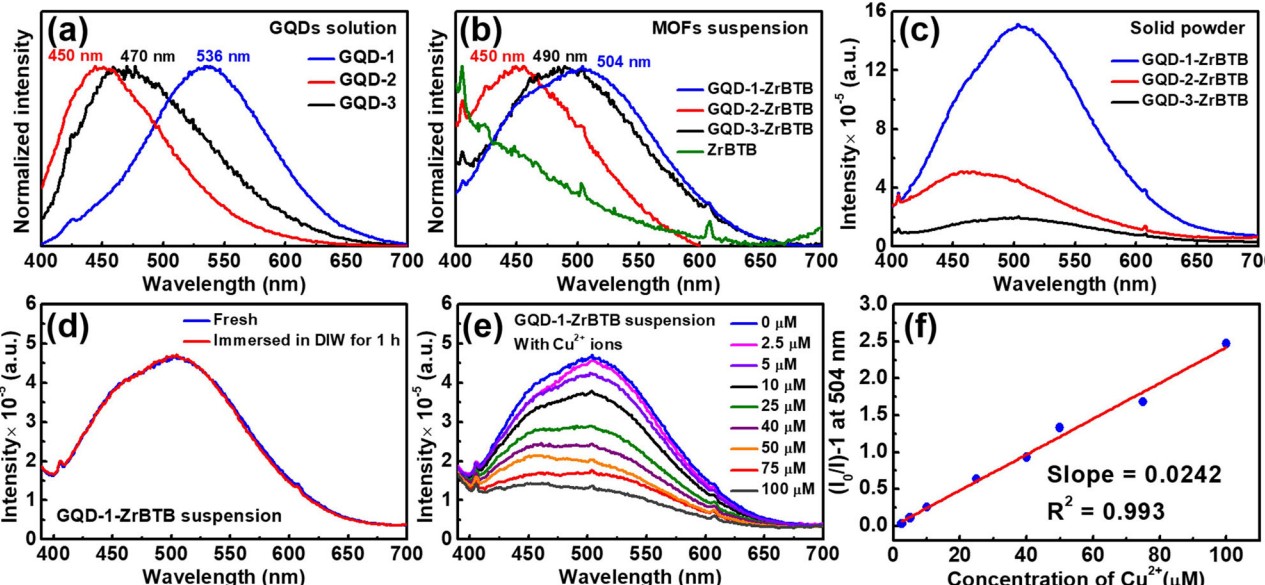

**Fig. 4 | PL spectroscopic data and sensing results.** Normalized emission spectra of (**a**) aqueous solutions of GQDs at 0.067 mg/mL and (**b**) suspensions of MOF-based materials in deionized water (DIW) at 0.4 mg/mL. Wavelengths of maximum emission are also marked. **c** Emission spectra of MOF-GQD composites in the form of dry solid powder. Emission spectra of GQD-1-ZrBTB dispersed in DIW at 0.4 mg/ mL, collected (**d**) before and after keeping for 1 h and (**e**) with various concentrations of $Cu^{2+}$. **f** Plot of $(I_0/I)$-1 versus the concentration of $Cu^{2+}$, extracted from the data shown in (**e**). $I$: emission intensity; $I_0$: emission intensity recorded without adding $Cu^{2+}$. All spectra were collected under the excitation at 370 nm.

response towards $Cu^{2+}$ ions compared to the suspension of GQD-1-ZrBTB, as mentioned in the introduction, GQDs immobilized in a solid MOF support are much easier for the separation and recycling after use. The selectivity of GQD-1-ZrBTB for ion sensing was also tested in the presence of various interferents. As revealed in Figure S21, the PL quenching of GQD-1-ZrBTB is highly selective toward $Cu^{2+}$ ions against other common metal ions except cobalt ions. In addition, the crystallinity of the MOF can be well preserved after the exposure to the aqueous solution containing $Cu^{2+}$ for 30 min (Fig. S22). The GQD-1-ZrBTB after the exposure to $Cu^{2+}$ ions was further re-dispersed in water, and the obtained suspension was subjected to PL measurements. As shown in Fig. S23, the PL response of the resulting suspension is almost the same as that measured in the presence of $Cu^{2+}$ ions. This GQD-1-ZrBTB material after use was further subjected to ICP-OES measurements, and a loading of around 0.2 copper atom per hexa-zirconium node was found in this used sample. Findings here suggest that the PL quenching of GQD-1-ZrBTB during the sensing process is mainly attributed to the adsorption of copper ions in GQD-1-ZrBTB.

## Conclusions
In summary, a water-stable 2D Zr-MOF, ZrBTB, can be used as the platform for the post-synthetic immobilization of GQDs. By taking advantage of the nature of 2D MOFs that can be fully dispersed as spatially separated molecular sheets as well as the chemical functional groups present in ZrBTB, GQDs with three distinct average sizes, all larger than the pore size of the ZrBTB sheet, can be immobilized in the 2D MOF. Installed GQDs spatially dispersed on the 2D MOF sheet can be observed. Unlike the pristine GQDs that can only show luminescent properties in the form of aqueous solution, the MOF-immobilized GQDs can still reveal photoluminescence in the form of dry solid. The synthetic approach proposed here should be generalizable to immobilize other pre-synthesized GQDs with various sizes and functionalities, not limited to the three GQD materials demonstrated here, in the ZrBTB framework.

## Material and methods
Experimental details for the synthesis of GQD-1, GQD-2, and GQD-3 and the preparation of their solutions can be found in Supplementary Methods. ZrBTB was synthesized solvothermally by using benzoaic acid as the

modulator, and the coordinated benzoate ligands were removed from the hexa-zirconium nodes of the MOF by an acidic treatment in a HCl/dimethyl sulfoxide mixture in order to obtain the accessible –OH/–OH$_2$ ligands on the 2D MOF sheets that can be used for further PSM[37,56]. The detailed synthetic procedure of ZrBTB can be found in our recently published work[36].

For the immobilization of graphene quantum dots with different sizes on ZrBTB molecular sheets, PSM was performed at room temperature. 30 mg of ZrBTB powder was homogeneously dispersed in 2.5 mL of the GQD-1, GQD-2, or GQD-3 solution (10 mg/mL) by ultrasonication, and the obtained suspension was then kept at room temperature for 24 h with periodically shaking in between. Thereafter, the resulting MOF solid was separated from the GQD-containing supernatant by centrifugation, and the solid was washed with 2.5 mL of water three times through centrifugation. Between each addition of water and the following centrifugation, the mixture was sonicated for 1 min in order to completely remove the uncoordinated GQDs from the MOF-based solid. Thereafter, 2.5 mL of acetone was added to disperse the obtained solid, and solvent exchange with acetone was conducted for three times over the course of overnight. After the activation of the MOF under vacuum at 60 °C overnight, the solid powder of GQD-1-ZrBTB, GQD-2-ZrBTB, or GQD-3-ZrBTB was obtained.

Other details regarding the instrumentation and PL measurements can be found in Supplementary Methods.

## Data availability
The data that support the findings of this study are available from the corresponding author upon reasonable request.

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

## Acknowledgments

We thank National Science and Technology Council (NSTC) of Taiwan for funding this work, under projects 112-2223-E-006-003-MY3, 112-2811-E-011-018-MY2, 111-2223-E-011-002-MY3, 111-2628-E-011-002-MY2, and 109-2923-E-011-003-MY3. This study was also in part supported by Ministry of Education (MOE) of Taiwan, under Yushan Young Scholar Program and SEED Project. We thank Ms. Chia-Ying Chien at Instrumentation Center of National Taiwan University (NTU) for FE-TEM experiments (JEOL JEM-2100F) with the financial support from NSTC. We also acknowledge Ms. Ju-Hung Ho at the Precious Instrumentation Center of National Taiwan University of Science and Technology for XPS experiments (PHI 5000 VersaProbe III). We also appreciate the support from MOE and NCKU under Higher Education Sprout Project. XPS data were in part collected by Surface Analysis Lab in Department of Chemical Engineering, NTU. We also acknowledge Core Facility Center of NCKU for HAADF-STEM experiments.

## Author contributions

Y.-L. Chen carried out the data curation, methodology, formal analysis, validation, investigation, and writing – original draft. D. Kurniawan was involved in the methodology, formal analysis, validation, and writing – original draft. M.-D. Tsai was involved in the data curation, validation, and formal analysis. J.-W. Chang, Y.-N. Chang, and S.-C. Yang contributed to the data curation and validation. W.-H. Chiang contributed to the funding acquisition, resources, and writing – review and editing. C.-W. Kung contributed to the conceptualization, funding acquisition, resources, supervision, validation, writing – original draft, and writing – review & editing.

## Competing interests

The authors declare no competing interests.
