## [Peer review file · Communications Chemistry]

Reviewers' comments:

Reviewer #1 (Remarks to the Author):

The submitted manuscript, "Two-dimensional metal-organic framework for post-synthetic immobilization of graphene quantum dots toward photoluminescent sensing" demonstrated that MoF-immobilized GQDs are stable in water and show fluorescence in liquid and solid states. The investigated aspect to avoid aggregation-induced fluorescence quenching is interesting and may have several applications. Detailed structural, microstructural, and compositional characterizations have been carried out to analyze the MoF-GQDs composite. In particular, the author carried out detailed STEM-HAADF imaging to confirm the formation of the MoF-GQDs hybrid at the nanoscale.

While the adopted synthesis method and observing optical emission in different forms of optical properties are noteworthy, major scientific discussion is lacking in the present form of the manuscript. One of the main drawbacks of this work is the lack of a mechanism behind the optical properties of immobilized GQDs in solid forms. I recommend this work for publication after a major revision.

Please find some of my specific comments below:

1. In the demonstrated approach, the author claimed that the immobilization of GQDs of 3-5 nm is achieved by post-synthetic integration with MoF. They also claimed that the pore size does not impose the spatial limitation in MoF. In that case, what is the purpose of MoF? Such functionalization could be done by other means as well.
2. Is there any specificity with respect to MoF to achieve the immobilization of GQDs? The author reported the MoF pore size of 1.2 nm while the GQDs are 3-5 nm; it would have been more interesting if the design of MoF with specific types of ligands enabled the immobilization of GQDs instead of random decorations.
3. In Figure, HAADF-STEM images of (a) GQD-1-ZrBTB, (b) GQD-2-ZrBTB, and (c) GQD-160 3-ZrBTB are presented in different magnifications, and especially the (c) GQD-160 3-ZrBTB could be compared in same magnification. The image and the schematic shown in Figures 3d and 3e indicate that the structure of MoF and GQDs immobilization are not correlated. In that case, it is unclear what makes the GQDs stable without aggregation.
4. In Figure 4, all samples' emission spectra are compared in suspension and solid forms. While the GQD-1-ZrBTB, GQD-2-ZrBTB, and GQD-160 3-ZrBTB show similar intensities in MoF suspension form, the intensity of GQD-2-ZrBTB, and GQD-160 3-ZrBTB are drastically decreased in solid form as shown in figure 4c. What is the reason for this? This aspect needs to be discussed in detail.

Reviewer #2 (Remarks to the Author):

The authors describe an interesting new strategy for the immobilization of graphene quantum dots within a two-dimensional MOF. The novelty of the work lies in:

- 1) A new postsynthetic approach towards anchoring quantum dots using coordination to available metal sites in a MOF support
- 2) The use of a 2D MOF as support for GQDs

However, the data supporting the execution of the strategy needs further evidence and interpretation to fully support the authors' claims. I recommend the following revisions before acceptance.

- 1) The authors do not provide a satisfactory synthetic procedure for the MOF, and only refer to a previous publication. The removal of coordinated benzoic acid by the acid washing procedure outlined

should be confirmed using digestion ¹H NMR experiments.

2) The FTIR carbonyl stretch for GQD-2 incorporation by bonding with uncoordinated carboxylates is much weaker than expected based on the uncoordinated carboxylate stretches in the GQD-1 and GQD-3 composites, suggesting some degree of physical intercalation, and consequently that the washing procedure may need to be optimized.

3) Despite the authors' explanation of the relative electron withdrawing abilities of different GQDs, it is not clear why the max. XPS peak for the GQD-3 incorporated Zr-btb is lower than even the unmodified Zr-btb MOF. This would suggest that GQD-3 binding results in overall

4) The BET surface area for unmodified Zr-BTB reported in refs. 39 and 40 is significantly higher than the value reported here: 425 m²/g vs. 310 m²/g – suggesting improper activation of the material. This may be partly responsible for the observed increase in surface area upon GQD incorporation rather than the inconsistent layer expansion effects observed between GQD-1, GQD-2, and GQD-3, and should be investigated.

5) The method details for analytical techniques are sometimes a single line and do not include any of the relevant experimental parameters to enable replication of the work. They must be elaborated as appropriate.

6) The mass fraction of incorporated GQDs is not reported – this is a key piece of information in evaluating the extent to which GQD is truly incorporated between stacked layers vs. whether it is only able to bind to the crystal surface.

7) Some evidence supporting the homogenous dispersion (e.g. Tyndall scattering) of Zr-btb in GQD solutions should be given.

Reviewer #3 (Remarks to the Author):

Graphene quantum dots immobilized in MOF are reported in this work, which are water stable. Photoluminescence measurements were carried out of these composites, and the detection of Cu ion is demonstrated at the end. Overall, I found the work seems to be more routine work and needs to be improved. The following suggestions (but not limited to) may help the authors to improve the manuscript.

1) I found it difficult to understand that the size of GQD increases from GQD1 to GQD3 which does not reflect from the photoluminescence spectra. For example, the smallest QD should have the largest bandgap (therefore, blue PL), but the result is the opposite. Although GQD3 somehow comes in between. Can the authors explain it? What are the corresponding absorption spectra?

2) Although the authors normalized the PL spectra, I found it very important to understand and compare the absolute PL intensity when GQDs are immobilized into MOFs and the pure GQDs. I believe there could be PL quenching which may have GQD size dependence. If so, the authors need to understand the reason in terms of electron/energy transfer or by other means. Time-resolved PL measurements may help to understand the situation better.

3) There is obvious PL quenching of GQD1/ZrBTB in the presence of Cu ions. What is interesting to see what happens in the presence of individual GQD1 and ZrBTB also. In addition, the quenching mechanism needs to be addressed.

4) No obvious change is noticed in the XRD pattern before and after GQD immobilization. Perhaps, Raman measurement may help.

5) There are a lot of curves presented in Fig. 2b, however, only four of them are leveled.

Referee: 1

Comments:

The submitted manuscript, "Two-dimensional metal–organic framework for post-synthetic immobilization of graphene quantum dots toward photoluminescent sensing" demonstrated that MoF-immobilized GQDs are stable in water and show fluorescence in liquid and solid states. The investigated aspect to avoid aggregation-induced fluorescence quenching is interesting and may have several applications. Detailed structural, microstructural, and compositional characterizations have been carried out to analyze the MoF-GQDs composite. In particular, the author carried out detailed STEM-HAADF imaging to confirm the formation of the MoF-GQDs hybrid at the nanoscale.

While the adopted synthesis method and observing optical emission in different forms of optical properties are noteworthy, major scientific discussion is lacking in the present form of the manuscript. One of the main drawbacks of this work is the lack of a mechanism behind the optical properties of immobilized GQDs in solid forms. I recommend this work for publication after a major revision.

Response: Thanks for the comments. The responses to each specific comment are attached below.

1. In the demonstrated approach, the author claimed that the immobilization of GQDs of 3-5 nm is achieved by post-synthetic integration with MoF. They also claimed that the pore size does not impose the spatial limitation in MoF. In that case, what is the purpose of MoF? Such functionalization could be done by other means as well.

Response: Thanks for the comment. The unique features and advantages of MOFs that motivated us to use them to immobilize GQDs rather than using other supports have been clearly stated in the introduction in the *original manuscript* with cited references. Please check the second paragraph of the introduction in **page 2** of the *revised manuscript*. In addition, the high density of terminal -OH/-OH₂ ligands present on every hexa-zirconium node of the 2D MOF selected here are the key to chemically immobilize GQDs; this point has been clearly stated in the *original manuscript* as well. Please check the paragraph starting in **page 3** of the *revised manuscript*. Without such a specific chemical functionality uniformly present on the entire 2D support, the chemical functionalization of GQDs cannot be done.

2. Is there any specificity with respect to MoF to achieve the immobilization of GQDs? The author reported the MoF pore size of 1.2 nm while the GQDs are 3-5 nm; it would have been more interesting if the design of MoF with specific types of ligands enabled the immobilization of GQDs instead of random decorations.

Response: Thanks for the comment, but the immobilization of GQDs reported here is not random decoration. Instead, GQDs are coordinated on the specific types of ligands. For GQD-1 and GQD-3, they are coordinated to the terminal -OH/-OH₂ ligands present on every hexa-zirconium node of the 2D MOF; for GQD-2, they are coordinated on the -COOH groups of the MOF linkers that are not fully coordinated by the nodes. These concepts have been clearly stated in the last paragraph of the introduction of the *original manuscript* and have been experimentally validated by FTIR and XPS analysis, also in the *original manuscript*. To make this point even more clear, the following sentences were further added into the *revised manuscript*.

“It is worth mentioning that ZrBTB was synthesized with the use of benzoic acid as the modulator, and the benzoate-coordinated 2D MOF, ZrBTB-BA, was further treated with hydrochloric acid to remove all capped benzoate ligands,³⁶ as evidenced by the ¹H nuclear magnetic resonance (NMR) data shown in Figure S6. After the complete removal of benzoate, terminal –OH/–OH₂ ligands become accessible on each node of the resulting ZrBTB, which are expected for the coordination of GQDs containing carboxylic groups on their surface” (starting from the end of page 4)

3. In Figure, HAADF-STEM images of (a) GQD-1-ZrBTB, (b) GQD-2-ZrBTB, and (c) GQD-160 3-ZrBTB are presented in different magnifications, and especially the (c) GQD-160 3-ZrBTB could be compared in same magnification. The image and the schematic shown in Figures 3d and 3e indicate that the structure of MoF and GQDs immobilization are not correlated. In that case, it is unclear what makes the GQDs stable without aggregation.

Response: Thanks for the comments. We have revised our **Figure 3** in the *revised manuscript*, and the current version shows HAADF-STEM images of the three materials at the same magnification.

In addition, we have added the following sentences in the *revised manuscript* to explain why GQDs are not spatially correlated with the lattice of the MOF exactly in the STEM image and comment on the potential aggregation of GQDs.

“It is worth noting that since all GQDs have much larger sizes compared to the aperture size of the 2D MOF and all three kinds of GQDs have their own ranges of particle size distributions (see Figure S3), it is not viable to observe the immobilized GQDs exactly correlated to the lattice fringes of ZrBTB. From Figure 3(a-c), some aggregations of GQDs on the 2D MOF sheet can also be found, especially more obvious in GQD-3-ZrBTB.” (starting from the end of page 8)

We agree that some GQDs immobilized in the MOF may still have certain degrees of aggregation, but the immobilization of GQDs through forming chemical bonds between GQDs and the 2D ZrBTB (as mentioned several times in the manuscript as well as mentioned in the responses to the previous two comments) could prevent the serious aggregation of GQDs after the immobilization. We also performed time-resolved PL experiments to support this point, and the results are included as **Figure S19** in the *revised supporting information*. The following discussions have been also included in the *revised manuscript*.

“Time-resolved PL measurements were also performed for all GQD solutions and suspensions of MOF-GQD materials. As shown in Figure S19, the suspension of each MOF-GQD composite only shows a minor decrease in the PL lifetime, even compared with the homogeneous solution of the corresponding GQDs. Findings here suggest that with the chemical immobilization of GQDs on the 2D MOF sheets, the aggregation of GQDs in the resulting MOF-GQD composites is not quite significant.” (page 11)

4. In Figure 4, all samples' emission spectra are compared in suspension and solid forms. While the GQD-1-ZrBTB, GQD-2-ZrBTB, and GQD-160 3-ZrBTB show similar intensities in MoF suspension form, the intensity of GQD-2-ZrBTB, and GQD-160 3-ZrBTB are drastically decreased in solid form as shown in figure 4c. What is the reason for this? This aspect needs to be discussed in detail.

Response: Thanks for the comment, but the GQD-1-ZrBTB, GQD-2-ZrBTB, and GQD-3-ZrBTB do not show similar intensities in the suspension form. The PL spectra of all three MOF suspensions and all the three GQD solutions are shown in **Figure S18** of the *revised supporting information*, which show the same trend in the PL intensity, not only to each other but also to the solid spectra shown in Figure 4(c). Figure 4(a) and Figure 4(b) are “normalized” spectra of Figure S18, which are convenient for further discussions on the peak shift. These descriptions and discussions have been already included in detail in the *original manuscript*. Please check the two paragraphs starting from the end of **page 9** and **page 10** of the *revised manuscript*.

Referee: 2

Comments:

The authors describe an interesting new strategy for the immobilization of graphene quantum dots within a two-dimensional MOF. The novelty of the work lies in:

- 1) A new postsynthetic approach towards anchoring quantum dots using coordination to available metal sites in a MOF support
- 2) The use of a 2D MOF as support for GQDs

However, the data supporting the execution of the strategy needs further evidence and interpretation to fully support the authors' claims. I recommend the following revisions before acceptance.

Response: Thanks for the positive comments. The responses to each specific comment are attached below.

1. The authors do not provide a satisfactory synthetic procedure for the MOF, and only refer to a previous publication. The removal of coordinated benzoic acid by the acid washing procedure outlined should be confirmed using digestion ^1H NMR experiments.

Response: Thanks for the comment and suggestion. The synthetic procedure of the same MOF is completely the same as that reported in our previous work, thus we have clearly indicated which literature to follow to repeat the synthesis in the *revised manuscript*. We prefer not to copy-paste the same procedure again in this paper. The readers are suggested to find the procedure in the literature.

“The detailed synthetic procedure of ZrBTB can be found in our recently published work.³⁶” (*page 14*)

We agree that confirming the effective removal of capped benzoate ligands is very important before performing the immobilization of GQDs. Thus, we have followed the suggestion to include the ^1H NMR data as **Figure S6** in the *revised supporting information*. The following sentences have been added into the *revised manuscript* regarding this point.

“It is worth mentioning that ZrBTB was synthesized with the use of benzoic acid as the modulator, and the benzoate-coordinated 2D MOF, ZrBTB-BA, was further treated with hydrochloric acid to remove all capped benzoate ligands,³⁶ as evidenced by the ^1H nuclear magnetic resonance (NMR) data shown in Figure S6. After the complete removal of benzoate, terminal $-\text{OH}/-\text{OH}_2$ ligands become accessible on each node of the resulting ZrBTB, which are expected for the coordination of GQDs containing carboxylic groups on their surface.” (*starting from the end of page 4*)

2. The FTIR carbonyl stretch for GQD-2 incorporation by bonding with uncoordinated carboxylates is much weaker than expected based on the uncoordinated carboxylate stretches in the GQD-1 and GQD-3 composites, suggesting some degree of physical intercalation, and consequently that the washing procedure may need to be optimized.

Response: Thanks for the comment, but the weaker signal of uncoordinated carboxylic acids in GQD-2-ZrBTB does not suggest any degree of physical intercalation. Instead, it indicates that the majority of GQD-2 should be immobilized on the uncoordinated carboxylic groups of linkers either present at structural defects in the 2D MOF sheet or on the edge of the sheet. The appearance of small peak associated with the linkage between the carboxylate group of ZrBTB and the amino group of GQD-2 also supports the chemical immobilization of GQD-2 in the MOF. These discussions have been already made in the *original manuscript*; please check the detailed discussions in the FTIR paragraph in **page 7** of the *revised manuscript*.

To test the binding between the immobilized GQD-2 and MOF, the GQD-2-ZrBTB was further washed with water, and the resulting supernatant after centrifugation was checked; no emission from GQD can be observed from this supernatant, as shown in **Figure S10** of the *revised supporting information*. Thus, the washing procedure after the synthesis does not need further improvements. The following sentence has been added in the *revised manuscript*.

“Even though the majority of GQD-2 are not coordinated on the hexa-zirconium nodes of ZrBTB, the immobilization is still fairly firm, as evidenced by the photoluminescent data shown in Figure S10.” (*page 7*)

3. Despite the authors' explanation of the relative electron withdrawing abilities of different GQDs, it is not clear why the max. XPS peak for the GQD-3 incorporated Zr-btb is lower than even the unmodified Zr-btb MOF. This would suggest that GQD-3 binding results in overall

Response: Thanks for the comment, but the XPS peak of GQD-3-ZrBTB is not lower than the unmodified ZrBTB. Please check the XPS data shown in Figure 2(c). The XPS peak of GQD-3-ZrBTB is more positive than the ZrBTB, suggesting that the immobilization of GQD-3 makes the Zr atoms in hexa-zirconium nodes more electron-deficient. This trend agrees well with the PL observations discussed later. These discussions have been already included in the *original manuscript*. Please check our detailed discussions in the XPS paragraph starting from **page 6** of the *revised manuscript*.

4. The BET surface area for unmodified Zr-BTB reported in refs. 39 and 40 is significantly higher than the value reported here: 425 m²/g vs. 310 m²/g – suggesting improper activation of the material. This may be partly responsible for the observed increase in surface area upon GQD incorporation rather than the inconsistent layer expansion effects observed between GQD-1, GQD-2, and GQD-3, and should be investigated.

Response: Thanks for the comment. Please note that ZrBTB is a 2D MOF, not a 3D rigid MOF, and its BET surface areas are known to vary a lot depending on the stacking of these 2D sheets, and the stacking of these 2D sheets is significantly affected by the final solvent used before the activation (please see: *J. Am. Chem. Soc.* 2023, 145, 49, 26580–26591; *Nat. Commun.* 10, 2911 (2019)). The BET surface areas of ZrBTB can vary

from around 220 m²/g to almost 500 m²/g. Thus, the slight difference in BET surface area between 310 m²/g (this study), 330 m²/g (Ref. 37), and 425 m²/g (Ref. 40) does not mean the improper activation of the MOF pore, but is attributed to the difference in sheet stacking. Thus, the increase in BET surface area after immobilizing GQD-1 should be attributed to the difference in the stacking and interlayered space between 2D MOF sheets. The increase in BET surface area after incorporating other guest ligands has been observed in previous studies (Ref. 37 & 40) as well, which has been well documented and discussed in detail in the *original manuscript*. The following sentence was further added into the *revised manuscript* with more cited references to support in order to further clarify this concept.

“It should be noted that owing to the 2D nature of ZrBTB, the solvent used during the activation of the MOF can significantly affect the stacking of these 2D MOF sheets and therefore the resulting BET surface area;⁴¹⁻⁴² its reported BET surface areas range from 220 to 450 m²/g depending on the final solvent used before the activation.^{37, 40, 42-43}” (page 5)

5. The method details for analytical techniques are sometimes a single line and do not include any of the relevant experimental parameters to enable replication of the work. They must be elaborated as appropriate.

Response: Thanks for the comment and suggestion. We have added more experimental details regarding all measurements in the *revised supporting information* as follows.

“All PL data were collected at a scan rate of 10 nm/s, with the sampling width of 1 nm and integration time of 0.1 s. Excitation and emission slits are both 2 nm for all PL measurements. Cuvette with the size of 1 cm×1 cm×3.5 cm was used for all PL measurements of solutions and suspensions.”

“Time-resolved PL (TRPL) spectra were recorded by using a HORIBA iHR320 emission mono spectrometer equipped with an amplifier and discriminator module (HORIBA CFD-2G), a laser diode with a peak wavelength of 371 nm (DeltaDiode DD-375L, HORIBA) connected to a diode conditioner (DC-N15-370-10, HORIBA) and a picosecond diode controller (DeltaDiode DD-C1, HORIBA), and a high throughput time-correlated single-photon counting controller (DeltaHub DH-HT, HORIBA). The emission used to record the PL lifetime was adjusted according to the emission peak of each sample under the excitation at 371 nm.”

“Raman measurement was performed at room temperature by using a JASCO 5100 spectrometer with a laser excitation wavelength and power of 532 nm and 4 mW, respectively. For preparing Raman samples, the GQD solution was drop-cast on a Si wafer and dried on a hotplate at 50 °C, while the MOF-based solid was spread and compressed on a glass slide. A Si wafer with a Raman shift of 520 cm⁻¹ was used to calibrate the spectrometer prior to all measurements.”

“..., with the wavelength of 1.54 Å, voltage of 40 kV, and current of 20 mA.”

“Around 100 mg of KBr and 0.5 mg of the MOF-based material were grinded together, and the obtained mixture was pelletized into a pellet for FTIR measurements.”

“Every sample was degassed at 110 °C for 4 h before collecting the isotherm.”

“Absorption spectra were measured with the use of an UV-2600 (Shimadzu).”

(pages S4-S7)

6. The mass fraction of incorporated GQDs is not reported – this is a key piece of information in evaluating the extent to which GQD is truly incorporated between stacked layers vs. whether it is only able to bind to the crystal surface.

Response: Thanks for the comment, but the mass fractions of GQDs were already reported in the *original manuscript*. Please check the ICP-OES results and discussions as well as the experimental details, at the beginning of **page 8** of the *revised manuscript* and in **page S17** of the *revised supporting information*.

Fractions of GQDs in all three composites are larger than 10 wt%, suggesting that GQDs are immobilized between the 2D MOF sheets rather than only on the external crystal surface of stacked MOF sheets. The following sentence was further added in the *revised manuscript* to clarify this point.

“Mass fractions of GQDs in all the three composites are higher than 10 wt%, which implies that GQDs are immobilized between the stacked 2D sheets of ZrBTB rather than solely attached on the external crystal surface of stacked 2D MOF sheets.” (*page 8*)

7. Some evidence supporting the homogenous dispersion (e.g. Tyndall scattering) of Zr-btb in GQD solutions should be given.

Response: Thanks for the suggestion. The testing results were included as **Figure S7** in the *revised supporting information*. The following sentence was added in the *revised manuscript*.

“By fully dispersing the 2D ZrBTB in solutions of the three GQDs (see Figure S7), GQDs could be immobilized on the 2D molecular sheet of ZrBTB.” (*at the beginning of page 5*)

The following sentence was added in the *revised supporting information*.

“Tyndall scattering can be well observed after dispersing ZrBTB in all the three GQD solutions by sonication, as revealed in Figure S7.” (*page S13*)

Referee: 3

Comments:

Graphene quantum dots immobilized in MOF are reported in this work, which are water stable. Photoluminescence measurements were carried out of these composites, and the detection of Cu ion is demonstrated at the end. Overall, I found the work seems to be more routine work and needs to be improved. The following suggestions (but not limited to) may help the authors to improve the manuscript.

Response: Thanks for the comments and suggestions. We have conducted extra experiments and analysis to further improve our work. The responses to each specific comment are attached below.

1. I found it difficult to understand that the size of GQD increases from GQD1 to GQD3 which does not reflect from the photoluminescence spectra. For example, the smallest QD should have the largest bandgap (therefore, blue PL), but the result is the opposite. Although GQD3 somehow comes in between. Can the authors explain it? What are the corresponding absorption spectra?

Response: Thanks for the comment and constructive suggestions. We agree with the reviewer that the smallest QDs should exhibit the shortest emission wavelength due to the quantum confinement effect. However, the

presence of surface functionalities like oxygen-containing functional groups and heteroatom dopants can also affect the PL of GQDs. The chemical functionalities of these three GQDs (obtained from XPS data in Figure S2) have been summarized in **Table S1** of the *revised supporting information*. The following sentences have been added into the *revised manuscript* to make a deep discussion on the peak locations and peak intensities of these GQDs.

“Although the sizes of GQDs follow the order of GQD-3>GQD-2>GQD-1, the wavelengths of maximum emission of these GQD solutions do not follow this order. It has been reported that a higher content of oxygen-containing groups can cause a red-shift in the PL emission of GQDs.⁵⁰⁻⁵¹ Since GQD-1 possesses an oxygen-to-carbon ratio of 1.46, which is much higher than those of GQD-2 (0.42) and GQD-3 (0.32) (see Table S1), despite having the smallest average particle size, GQD-1 exhibits the longest emission wavelength among all the three kinds of GQDs. On the other hand, doping graphitic nitrogen into GQDs can result in a red-shift PL behavior, while both pyridinic nitrogen and pyrrolic nitrogen dopants can induce a blue shift in the PL emission.⁵⁰ From Figure S2d, it can be observed that GQD-2 contains various kinds of nitrogen dopants except graphitic nitrogen, leading to the strong blue shift in its emission spectrum. These results imply that surface functionalities could provide a stronger impact in the PL emission of GQDs than the particle size, especially when the difference in particle size is relatively small, *i.e.*, around 1-2 nm. In addition, Figure S18(a) also reveals that under the same concentration, the emission intensities of GQD solutions follow the order of GQD-1>GQD-2>GQD-3, which is consistent with the order of their oxygen-to-carbon ratios listed in Table S1. This observation should be attributed to the fact that a high content of surface functionalities in GQDs can prevent the π - π stacking-induced photon reabsorption and nonradiative energy transfer between GQDs, reducing the PL quenching phenomenon.⁵² After the immobilization of GQDs in ZrBTB, as shown in Figure S18(b), both the wavelengths of maximum emission and PL intensities in general follow the trend of GQD solutions, but some obvious peak shifts can be observed.” (*page 10*)

The following sentence was also added in the *revised supporting information* for Table S1.

“Elemental ratios obtained from the XPS data shown in Figure S2 are summarized in Table S1, revealing the significant difference in the chemical functionalities of the three GQDs.” (*page S9*)

The absorption spectra were also measured, and the data have been included as **Figure S16** in *revised supporting information*. The following sentences have been added in the *revised supporting information*.

“Absorption spectra were measured with the use of an UV-2600 (Shimadzu).” (*page S7*)

“Figure S16 shows the absorption spectra of the three GQD solutions, revealing a broad absorption band that corresponds to the presence of various electronic states. The absorption bands located at around 250-290 nm and 300-400 nm in each spectrum can be ascribed to the $\pi \rightarrow \pi^*$ transition of C=C bond and $n \rightarrow \pi^*$ transition of functional groups, respectively.¹⁶” (*page S22*)

2. Although the authors normalized the PL spectra, I found it very important to understand and compare the absolute PL intensity when GQDs are immobilized into MOFs and the pure GQDs. I believe there could be PL quenching which may have GQD size dependence. If so, the authors need to understand the reason in terms of electron/energy transfer or by other means. Time-resolved PL measurements may help to understand the situation better.

Response: Thanks for the comment, but all PL spectra before normalization have been already shown in the *original version of supporting information*. Please check **Figure S18** in the *revised supporting information*. The difference in absolute intensities in these spectra have been discussed in detail in the *revised manuscript*, as mentioned in the response to the previous comment#1.

Time-resolved PL data were added in the *revised supporting information* as **Figure S19**. The following sentences have been added into the *revised manuscript*.

“Time-resolved PL measurements were also performed for all GQD solutions and suspensions of MOF-GQD materials. As shown in Figure S19, the suspension of each MOF-GQD composite only shows a minor decrease in the PL lifetime, even compared with the homogeneous solution of the corresponding GQDs. Findings here suggest that with the chemical immobilization of GQDs on the 2D MOF sheets, the aggregation of GQDs in the resulting MOF-GQD composites is not quite significant.” (page 11)

3. There is obvious PL quenching of GQD1/ZrBTB in the presence of Cu ions. What is interesting to see what happens in the presence of individual GQD1 and ZrBTB also. In addition, the quenching mechanism needs to be addressed.

Response: Thanks for the comment and suggestion. The sensing data with homogeneous system of GQD-1 and solely with ZrBTB were added as **Figure S20** in the *revised supporting information*, and the following sentences have been added into the *revised manuscript*.

“The suspension of the pristine ZrBTB and the homogeneous solution of GQD-1 were also subjected to the PL detection of Cu²⁺ ions. As shown in Figure S20, no sensing response can be observed for the pristine ZrBTB, while the PL intensity of the GQD-1 solution can be significantly quenched after adding Cu²⁺ ions. This finding indicates that GQD-1 is the active material responsible for the PL sensing of Cu²⁺.” (starting from the end of page 12)

We also tried to characterize the recycled GQD-1-ZrBTB to investigate the possible sensing mechanism. The results including **Figure S23** in the *revised supporting information* and the following sentences in the *revised manuscript* were added.

“The GQD-1-ZrBTB after the exposure to Cu²⁺ ions was further re-dispersed in water, and the obtained suspension was subjected to PL measurements. As shown in Figure S23, the PL response of the resulting suspension is almost the same as that measured in the presence of Cu²⁺ ions. This GQD-1-ZrBTB material after use was further subjected to ICP-OES measurements, and a loading of around 0.2 copper atom per hexa-zirconium node was found in this used sample. Findings here suggest that the PL quenching of GQD-1-ZrBTB during the sensing process is mainly attributed to the adsorption of copper ions in GQD-1-ZrBTB.” (page 12)

4. No obvious change is noticed in the XRD pattern before and after GQD immobilization. Perhaps, Raman measurement may help.

Response: Thanks for the comment and suggestion. We agree that the change in PXRD pattern after GQD immobilization is not quite obvious, as already discussed in **page 5** of the *revised manuscript*. We have tried to measure Raman spectra, but we could not get much new information from the data. The data are now included as **Figure S11** in the *revised supporting information*. The following sentences were also added.

“Raman spectroscopic experiments of all MOF-based materials were attempted to further characterize the immobilized GQDs, but it was not successful owing to the presence of strong Raman peaks from the pristine ZrBTB within the same region of the characteristic peaks of GQDs (Figure S11).” (*revised manuscript, page 7*)

“Raman measurement was performed at room temperature by using a JASCO 5100 spectrometer with a laser excitation wavelength and power of 532 nm and 4 mW, respectively. For preparing Raman samples, the GQD solution was drop-cast on a Si wafer and dried on a hotplate at 50 °C, while the MOF-based solid was spread and compressed on a glass slide. A Si wafer with a Raman shift of 520 cm⁻¹ was used to calibrate the spectrometer prior to all measurements.” (*revised supporting information, page S6*)

5. There are a lot of curves presented in Fig. 2b, however, only four of them are leveled.

Response: Thanks for the comment. We have labeled adsorption and desorption curves separately in the revised version of **Figure 2(b)**.

REVIEWERS' COMMENTS:

Reviewer #1 (Remarks to the Author):

The revised version may be considered for acceptance.

Reviewer #2 (Remarks to the Author):

I believe the revisions undertaken and responses provided by the authors are satisfactory and the manuscript may be accepted for publication.

Reviewer #3 (Remarks to the Author):

The authors have addressed most of my comments and I am happy to recommend it for publication. However, the following two points the author should consider again:

- 1) There is a clear quenching in the lifetime decay traces (in Fig. S19) which they mention as minor quenching: "...each MOF-GQD composite only shows a minor decrease in the PL lifetime, even compared with the homogeneous solution of the corresponding GQDs." This should be corrected, and the relevant discussion needs to be updated. The $\tau(\text{avg})$ in Fig. S19a for the composite seems overestimated (should be around 3 ns).
- 2) Since the PL intensity of the GQD-1 solution can be significantly quenched after adding Cu^{2+} ions, this would then still raise the question of why you need GQD1/ZrBTB for Cu ion detection if GQD1 itself can do that? It appears the quenching (sensing) is even more drastic in GQD1 alone.

Referee: 1

Comments:

The revised version may be considered for acceptance.

Response: Thanks for the positive feedback and we appreciate the effort from the reviewer in reviewing this manuscript.

Referee: 2

Comments:

I believe the revisions undertaken and responses provided by the authors are satisfactory and the manuscript may be accepted for publication.

Response: Thanks for the positive feedback and we appreciate the effort from the reviewer in reviewing this manuscript.

Referee: 3

Comments:

The authors have addressed most of my comments and I am happy to recommend it for publication. However, the following two points the author should consider again:

Response: Thanks for the positive comments. We have addressed the following two points in our *revised manuscript*, and we believe that with the revision this manuscript is ready for publication. We do appreciate the effort from the reviewer in reviewing this manuscript.

1) There is a clear quenching in the lifetime decay traces (in Fig. S19) which they mention as minor quenching: "...each MOF-GQD composite only shows a minor decrease in the PL lifetime, even compared with the homogeneous solution of the corresponding GQDs." This should be corrected, and the relevant discussion needs to be updated. The $\tau(\text{avg})$ in Fig. S19a for the composite seems overestimated (should be around 3 ns).

Response: Thanks for the comments. We have checked the $\tau(\text{avg})$ fitting for the suspension data shown in Fig. S19(a), and we found that the fitted $\tau(\text{avg})$ should be 3.16 ns, not 3.96 ns. The number has been corrected in **Figure S19(a)** of the *supplemental information*. We are sorry for the mistake.

We have corrected the sentences with more discussions included in the *revised manuscript* related to the TRPL results. Please check the following sentences.

"As shown in Figure S19, the suspensions of GQD-1-ZrBTB, GQD-2-ZrBTB and GQD-3-ZrBTB possess around 70%, 62% and 58% of the corresponding PL lifetimes of homogeneous GQD solutions, but all PL lifetimes are in the same order of magnitude within a few nanoseconds. Findings here suggest that with the chemical immobilization of GQDs on the 2D MOF sheets, the aggregation of more GQDs in the resulting MOF-GQD composites may occur compared to that in homogeneous solutions, but such an aggregation is not quite significant."

(page 9 of the revised manuscript)

2) Since the PL intensity of the GQD-1 solution can be significantly quenched after adding Cu^{2+} ions, this would then still raise the question of why you need GQD1/ZrBTB for Cu ion detection if GQD1 itself can do that? It appears the quenching (sensing) is even more drastic in GQD1 alone.

Response: Thanks for the comments. The advantage of MOF-immobilized GQDs compared to the GQD solution has been already mentioned in the first paragraph of the introduction. To further emphasize it in the Cu^{2+} -sensing part of the manuscript, the following sentence was further included in the *revised manuscript*.

“It is worth mentioning that even though the homogeneous solution of GQD-1 seems to show a better sensing response towards Cu^{2+} ions compared to the suspension of GQD-1-ZrBTB, as mentioned in the introduction, GQDs immobilized in a solid MOF support are much easier for the separation and recycling after use.”

(page 10 of the revised manuscript)